# Polycrystalline PbTe:In Films on Amorphous Substrate: Structure and Physical Properties

**DOI:** 10.3390/ma15238383

**Published:** 2022-11-25

**Authors:** Vadim Kovalyuk, Evgeniia Sheveleva, Mark Auslender, Gregory Goltsman, Roni Shneck, Zinovi Dashevsky

**Affiliations:** 1NTI Center for Quantum Communications, National University of Science and Technology MISiS, 119049 Moscow, Russia; 2Tikhonov Moscow Institute of Electronics and Mathematics, National Research University Higher School of Economics, 101000 Moscow, Russia; 3Department of Physics, Moscow State Pedagogical University, 119992 Moscow, Russia; 4School of Electrical and Computer Engineering, Ben-Gurion University, Beer-Sheva 84105, Israel; 5Department of Materials Engineering, Ben-Gurion University, Beer-Sheva 84105, Israel

**Keywords:** lead chalcogenide, polycrystalline films, doping of indium, transport properties, barrier scattering

## Abstract

Polycrystalline PbTe:In films on a polyimide substrate were obtained and investigated. Their structural and transport properties in a wide range of temperatures (10–300 K) were studied. The unique feature of In impurity in PbTe is the stabilization of the Fermi level (pinning effect) that allowed for the preparation polycrystalline films with the same carrier concentration. We found that heat treatment in an argon atmosphere does not change the average grain size and carrier concentration for as-grown films but greatly increases the Hall mobility and the electron mean free path. By comparing the mobility in the bulk and in the film after heat treatment, we extracted the value of the mobility that arises due to scattering at the grain boundary barriers. The ultimate goal of the present study is the development of these films in designing portable uncooled photodetectors for the mid-IR range.

## 1. Introduction

Lead telluride (PbTe), a narrow-bandgap semiconductor with an energy gap *E*_g_ varying from 190 meV at *T* = 0 K to 319 meV at room temperature, is commonly applied in devices for the IR spectral region [1,2,3,4,5,6,7,8]. During WWII, lead chalcogenides (PbS, PbSe and PbTe) found military applications for the detection of the infrared part of the spectra [9]. Infrared photodetectors have received much attention for several decades due to their broad applications in the military, science and daily life [1]. 

However, because of the narrow energy gap and lacking well trained reproducible routes of fabrication of high-quality photosensitive lead chalcogenides thin films, the development of portable detectors for the mid-IR range based on these materials is still a timely task. Different methods for fabricating lead chalcogenide thin films evolved rapidly, including vapor and chemical deposition techniques [10,11,12,13]. The n-type PbTe has also been widely used as a basis for thermoelectric generators [14,15].

For years, the main research on PbTe films was focused on the development of high-quality epitaxial PbTe films on different substrates (BaF_2_, NaCl, mica) [16]. The high value of electron mobility (µ_e_) at a level of 10^5^ cm^2^/Vs at temperature *T* = 4.2 K was demonstrated for the n-type PbTe films grown on BaF_2_ by hot wall epitaxy and MBE [10,11]. Such a result exceeds the previously shown electron mobility values for the PbTe single crystals grown by the Czochralski technique [17].

The first theory of high sensitivity in polycrystalline lead chalcogenide films after their heat treatment in the oxygen atmosphere was proposed by Petritz R.L. more than 60 years ago [18]. It suggests that the photoconductive films of the lead salt family are composed of a system of crystallites separated by intercrystallite barriers. The “crystallites” are represented by the lead salts, while a lead oxide or a lead oxide salt plays the role of the “Intercrystallite barriers” [19]. The space charge regions are localized at the surface of the crystallites. The lifetime of a hole-electron pair is partially determined by the surface states, while the resistivity is strongly affected by the intercrystallite barriers. Since then and until now, no extended experimental and theoretical investigations of polycrystalline lead chalcogenide films have been reported. 

A promising method to control the electron concentration of PbTe semiconductors is doping with indium, which forms quasi-local (resonant) levels in the conduction band [20,21,22]. Traditional impurities do not create shallow hydrogen-like levels in PbTe and related compounds, as these materials have a pronounced ionic component of the chemical bond and a high value of the static dielectric constant ε. In addition, these compounds contain a certain carrier concentration (up to 10^18^–10^19^ cm^−3^), which does not freeze out down to helium temperature [23]. Doping with indium brings some unique properties to lead telluride that are related to Fermi-level pinning [24,25]. 

The special feature of the films studied in this work is that the PbTe:In films were grown on a polyimide substrate (Kapton) [26]. The resulting films have low resistivity with degenerated electron statistics. Fermi level pinning makes similar concentrations for films of different doping levels, thereby excluding the contraction dependence of mobility, and high stability during thermal cycling. A large number of grains (serving as barriers) of PbTe:In on a polyimide substrate should suit ideally for studying the barrier scattering [19].

## 2. Materials and Methods

The synthesis of PbTe-based materials was carried out by direct melting of high-purity components for 10 h at 1073 K in the sealed quartz ampoules evacuated to a residual pressure of 10^−5^ mbar. Each ampoule was then taken from the furnace and quenched in cold water. The obtained ingots were crushed into fine powders by ball milling in an Ar atmosphere.

The n-type PbTe:In thin films on an amorphous substrate (the polyimide film, 100 µm thick) were deposited by electron-beam physical vapor deposition (EBPVD) through a rigid metal mask using Edwards E306A at vacuum ~10^−5^ mbar (Figure 1a).

The growth rate was controlled by a piezo sensor and maintained at the level of 0.1–0.2 nm/s. The films were grown to 2, 3 and 4 μm thickness at the substrate temperatures (*T_S_*) that varied from 523 to 638 K. We studied the films with two indium doping concentrations equal to 0.005, 0.01. Ohmic Ni contacts of Hall geometry were made through the second metal mask.

The post-growing processing of the films was carried out in a specially created vacuum-sealed stainless-steel case, placed in an oven. After pumping down to 10^−3^ mbar and Ar washing, the as-grown films were placed in an argon atmosphere at an excess pressure of 1 atm. The temperature of the oven was 673 K, and the annealing of the films lasted for 3 h. The whole cycle until complete cooling lasted 24 h.

X-ray Rigaku Model-2000 X-ray diffractometer with K-series α-line radiation of Cu atom λ = 1.5405 Å was used to study the structural properties of the films. 

Secondary emission electron images were taken with the high-resolution JEOL JSM-7400F HRSEM scanning electron microscope (SEM) with an accelerating voltage of 3 kV. The magnitude of the accelerating voltage is critical since the polyimide substrate might melt at high voltages.

For the investigation of the transport properties on thin films (electrical conductivity σ and Hall coefficient RH) over a wide temperature range of 10–300 K, we used an original experimental setup based on a closed cycle Gifford-McMahon cryocooler SRDK-101D (Figure 1b). The measurements of the Hall effect were performed in permanent magnetic fields up to 0.8 T. The results are combined from the averaged measurements in two directions of the electrical and magnetic fields. To reduce the thermal load and reduce the number of wires suitable for the sample in the cryostat, we used an electrical switch that allows to switch the multimeter between the Hall and the potential type of contacts. The lowest achieved temperature (10 K) was limited by the heat flow from the magnet manipulator. Temperature stabilization was achieved with the CryCon 32B controller. 

The accuracy of the temperature measurement was 0.1 K, and the magnetic field measurements ranged in ±3% stability. The uncertainty of the electrical conductivity measurements was 6%. The Hall effect was measured with an accuracy of 8%. 

The main technological parameters of PbTe:In are presented in Table 1.

## 3. Results

### 3.1. Structural Properties

Structural properties of the as-grown PbTe:In films, including the grain size evolution with increasing substrate temperature, are shown in Figure 2a–c. As the substrate temperature increases, the average grain size increases from 114 ± 11 nm at *T_S_* = 523 K to 510 ± 54 nm at *T*_S_ = 638 K. 

As the film thickness increases at a constant substrate temperature (*T_S_* = 598 K), the average grain size also increases from 300 ± 35 nm for a film thickness of 2 µm to 640 ± 70 nm for a film thickness of 4 µm (Figure 2b,g,h).

It is interesting to note that the average crystallite size after annealing (656 ± 50 nm) remains almost unchanged (641 ± 70 nm), as can be seen in Figure 2d,e,h.

All films grown on an amorphous substrate are characterized by a columnar structure, as shown in the example of a Pb_0.995_In_0.005_Te film of 3 µm thickness (*T_S_* = 623 K) in Figure 2i.

### 3.2. XRD

Figure 3a shows the XRD spectra corresponding to the films grown at different substrate temperatures *T_S_*. Similar to a single crystal with a cubic structure, the PbTe:In films tend to grow predominantly in the [100] direction. The main peaks for all the samples are several orders of magnitude larger than the additional ones, which proves that the films are of high quality. The peaks corresponding to the Pb, Sn and PbO phases were not found, which indicates the single-phase nature of the films. As the substrate temperature rises, the additional peaks corresponding to the growth planes (311) and (222) almost completely disappear. Heat treatment also leads to a great improvement in the film structure, which results in a thinning of the main peaks as well as in the disappearance of additional (311) and (222) growth planes (Figure 3b).

### 3.3. Seebeck Coefficient

The measured negative values of the Seebeck coefficient (thermopower) for the fabricated PbTe:In films indicate electron conductivity. The values of the Seebeck coefficients are in the range of −150 µV/K, corresponding to the electron concentration at the level of *n* = 4–5×10^18^ cm^−3^ at room temperature, which is close to the ones seen in PbTe:In bulk crystals [24].

### 3.4. Hall Coefficient and Energy Bands Diagram Reconstruction

The temperature dependence of the Hall coefficient in the range of 10–300 K for as-grown Pb_0.995_In_0.005_Te film is shown in Figure 4a (empty triangles). The Hall coefficient is negative, and its temperature trend from −0.6 cm^3^/C at 300 K to −0.3 cm^3^/C at 10 K is similar to the one observed in PbTe:In bulk crystals [18,19]. Despite the change in the In concentration of Pb_0.99_In_0.01_Te Figure 4a (empty squares), the behavior of the Hall coefficient practically does not change in the as-grown films, like in bulk materials. 

For the first time, a strong and nonmonotonic temperature dependence of the Hall coefficient, unusual for n-PbTe, was discovered in Ref. [27]. The dependence of the Hall coefficient on the concentration of In was explained by the presence of a level that occurs when In is introduced into PbTe. At low temperatures, it lies above the bottom of the conduction band and drops with increasing temperature into the forbidden band. Fermi energy increases with increasing content of In and reaches the quasilocal level, whereupon a further increase in the concentration of free electrons ceases, and the Fermi energy is stabilized near the impurity level. Upon shifting down with temperature, the level draws the Fermi level along with it. This leads to corresponding changes in the Hall concentration. Besides depressing the impurity level, increasing temperature increases the degree of its thermal ionization. Studies of transport phenomena performed by the authors of Refs. [16,28,29] have confirmed the data of Ref. [27] have filled out the details substantially.

Based on the known theoretical prediction of the conduction and valence bands, and the Fermi level positions in PbTe, we reconstructed the band diagram and added our experimental data on the carrier concentration (*n*) there, using the following semi-empirical formula [23] for the dependence of the bandgap on the temperature (*T*):(1)$$E_{g}\left(T\right)=0.19+0.45^{-3}\;\frac{T^{2}}{T+50}\;,$$
where *E*_g_ and *T* are in units of eV and degrees K, respectively.

The position of the Fermi level (chemical potential) ζ for In doped PbTe can be calculated using the following two models: the parabolic Kane’s model [30]:(2)$$n=M_{C}\frac{\sqrt{2}{\left(m_{d}^{*}k_{B}T\right)}^{3/2}}{\pi^{2}\hbar^{3}}F_{1/2}\left(\zeta^{*}\right),\;$$
and non-parabolic Cohen’s model [31]:(3)$$n=M_{C}\frac{\sqrt{2}{\left(m_{d}^{*}k_{B}T\right)}^{3/2}}{\pi^{2}\hbar^{3}}\left[F_{1/2}\left(\zeta^{*}\right)+2\frac{k_{B}T}{E_{g}}F_{3/2}\left(\zeta^{*}\right)\right]\;.$$

Here *n* is the electron concentration, *M*_C_ is the number of equivalent energy minima of the conduction band, *k*_B_ is the Boltzmann constant, *ħ* is the Planck constant, *F*_α_(*x*) are the Fermi integrals, ζ*= ζ / *k*_B_T is the reduced chemical potential, and *m*_d_^*^ is the density-of-states effective mass for the conduction band, which is given by *m*_d_* = (*m*_l_*∙ *m*_t_*^2^)^1/3^, *m*_l_* and *m*_t_* being the longitudinal and transverse electron effective masses. The relations between these masses and their expression via the bandgap are well known [23].

The reconstructed energy band diagram, showing the valence band ceiling (*E*_V_), the bottom line of the conduction band (*E*_C_) and the Fermi level (*E*_F_) line relative to the middle of the bandgap (*E*_i_) is plotted in Figure 4b. 

The positions of the Fermi level calculated via the parabolic Kane model and the non-parabolic Cohen model are shown in Figure 4b in empty red squares (line 2) and empty red circles (line 3), respectively. At temperatures above 150 K with weak degeneracy, both models are in good agreement with theoretical predictions, while at low temperatures and strong degeneracy, the nonparabolicity should be taken into account. The temperature dependence of the Hall coefficient for grown PbTe:In, as in the bulk material case corresponds to the variation with the temperature of the Fermi level to the conduction band edge separation. Due to the Fermi level pinning, this variation occurs mostly on account of the band edge variation, which is in good agreement with experimental data. The Fermi level pinning ensures that the electrical properties of large enough PbTe crystalline grains are similar, which therefore provides high reproducibility of these properties of the polycrystalline PbTe.

### 3.5. Mobility

Based on the measured temperature dependences of the Hall coefficient (*R*_H_) and conductivity σ, we determined the Hall mobility as follows: µ_H_(*T*) = *R*_H_σ, where σ is the film conductivity.

As the first step, we measured the effect of the substrate temperature (*T*_S_) on the mobility µ_H_ of the grown films. Figure 5 presents the measured temperature dependence trends of the Hall mobility of 2 µm thick Pb_0.99_In_0.01_Te grown at various substrate temperatures (*T*_S_ = 523–638 K). 

For all cases, the mobility function has the shape of a bell as the temperature decreases, it increases and then, after reaching a maximum, begins to fall. 

With an increase in the substrate temperature along with an increase in the grain size increases (Figure 1a–c) and the Hall mobility µ_H_ increases at room and helium temperatures. The maximum µ_H_ =1522 cm^2^/Vs is reached at *T*_S_ = 638 K, measured at a temperature of 120 K.

In the second step, the Hall mobility for the films of several thicknesses was measured. In Figure 6, measured temperature dependences of the Hall mobility of Pb_0.99_In_0.01_Te (fixed *T*_S_ = 598 K) for the following film thicknesses: 2 µm, 3 µm and 4 µm are shown. As the film thickness and the grain size increase (Figure 1b,g,h), the Hall mobility µ_H_ also increases. The temperature dependence of the mobility is of a bell shape again, but the maximum shifts towards lower temperatures from 140 K and maximum mobility of 1036 cm^2^/Vs (2 µm thickness) to 100 K and the maximum of µ_H_ reaches of 2120 cm^2^/Vs.

In the third step, we studied the Hall mobility of Pb_0.995_In_0.005_Te films before and after heat treatment. The heat treatment leads to a significant increase in the µ_H_ value. At helium temperatures, the mobility increases by more than an order of magnitude, from 1440 cm^2^/Vs to 18870 cm^2^/Vs (Figure 7). 

At room temperature, the Hall mobility of PbTe films approaches the µ_H_ of a bulk crystal, and its trend corresponds to electron scattering by phonons (~*T*^−2.5^) [23].

## 4. Discussion

We have obtained PbTe:In polycrystalline films with electron concentrations close to those of PbTe:In bulk materials. However, the carrier mobility in the films does not nearly reach the values observed in bulks, even after annealing in an argon atmosphere.

The difference between the transport of bulk single crystals and polycrystalline films is associated with the following three main factors: Scattering on the free surface;Scattering on crystalline boundaries;Scattering on defects and dislocations.

Sufficiently thick films were grown to exclude surface scattering. Using Matissen’s rule we can write the expression for the effective Hall mobility as follows:(4)$$\mu_{H}^{-1}=\mu_{L}^{-1}+\mu_{\mathrm{def}}^{-1}+\mu_{\mathrm{gb}}^{-1},$$
where μ_L_ is the Hall mobility due to lattice scattering in bulk crystal, μ_gb_ is the mobility limited by grain boundaries and μ_def_ is the mobility limited by the defects and dislocations. 

In a case of elastic scattering, the mobility is equal to μ = *e* ⟨τ⟩/*m**, where ⟨τ⟩ is the average relaxation time; *m** is the effective mass. In the simplest cases, the relaxation time is the power function of the carrier energy *E*, temperature *T* and effective mass *m** and is given by τ ∝ *E*^r^ *T*^s^ *m**^t^ [23]. For the acoustical vibrations of the lattice, one can show *r* = −1/2, *s* = −1, *t* = −3/2. It has now been established that the mobility in lead chalcogenides at high temperatures is described by the formula μ ∝ *m**^−5/2^
*T*^−3/2^. Given the fact that *m**∝ *T*^−1^, then μ ∝ *T*^−5/2^. Obtaining from the experimental dependences *r*_eff_ = dln(μ)/d*T* one can estimate how close the scattering in the film is to the scattering due to lattice vibrations in the bulk material. In as-grown films, the mobility at 300 K is less than in a bulk single crystal and has a scattering parameter *r*_eff_ = −1.6, which absolute value is smaller than in the bulk single crystals [13]. After annealing in argon mobility at high temperatures becomes close to the bulk single crystals and *r*_eff_ = −2.5 begins to correspond to the phonon scattering. Considering that the average grain size practically does not change (Figure 2d,e), but on the other hand, the crystallographic features changed close to those of bulk crystals (Figure 3b). Therefore, we associate the increased Hall mobility with the improved crystalline structure of the heat-treated PbTe:In films. 

In the case of degenerate statistics and anisotropy of energy ellipsoids, a formula was obtained for calculating the mean free path length [16]:(5)$$\lambda =\frac{\hbar }{e}\mu_{F}{\left(\frac{3}{4\pi^{2}n}\right)}^{\frac{1}{3}},$$
where e is the charge of an electron, μ_F_ is the mobility at the Fermi level and *n* is the electron concentration.

The mean free path calculated by Equation (5) at *T* = 10 K after heat treatment in argon changes from 42 nm to 515 nm. In this case, it becomes comparable with the size of crystallites (656 ± 50 nm). In the best PbTe single crystals, the Hall mobility is two orders of magnitude higher than that in films on a polyimide substrate [32]. This confirms the limitation of the mean free path in films due to the crystallite’s size.

By comparing the temperature dependence of the mobility for PbTe single crystals [32] and μ_H_ for measured films, we conclude that the Hall mobility of films is limited by defects μ_def_ and grain boundary scattering (μ_gb_) (Equation (4)). Due to the large difference in motility values before µ_b_ and after µ_a_ heat treatment (in order of magnitude), we can consider them separately to find the mobility restricted by defects μ_def_^−1^=μ_b_^−1^ − μ_L_^−1^, as well as mobility restricted by grain boundaries μ_gb_^−1^ = μ_a_^−1^ − μ_L_^−1^.

The first graph of the mobility µ_L_ is a classical dependence for a bulk crystal. At high temperatures, the dependence obeys the power-law expression μ ∝ *T*^−5/2^. At low-temperature saturation Hall mobilities in A^IV^–B^VI^ materials are entirely due to dislocations and are not caused by ionized impurity scattering (as in technically important diamond or zinc blende type semiconductors).

The mobility extracted from experimental data µ_def_ shows the contribution of defects for as-grown films to the mobility. It weakly depends on temperature, rising slightly closer to room temperature. The small mobility value is associated with a large number of defects and dislocations compared to bulk materials. 

Another mobility µ_gb_ extracted from the experimental data is associated with the barrier scattering mechanism. As can be seen from Figure 8a, at low temperatures, the mobility is almost temperature independent, which is consistent with the scattering by defects.

The obtained dependence on the mobility can be explained by the creation of localized electronic states at the inter-crystalline boundaries. When filling, the boundaries become charged, generating potential barriers for electron scattering [16]. An impurity indium atom (an element from the third group) replaces the Pb^2+^ ion in the crystal lattice site and donates an additional electron to the conduction band. It becomes an effective positive charge at the corresponding Pb lattice site, acquiring the In^3+^ charge state. Since the ionic radius *R*(In^3+^) = 0.09 nm is smaller than *R*(Pb^2+^) = 0.13 nm, indium ions accumulate in the compression region near the dislocation, while the extension region becomes depleted of indium. The signs of the deformation potential constants are such that a potential hump is formed in the compression region and a potential well is formed in the tension region. The combined effect of In doping is to reduce the potential barriers on the grain boundaries (formed from dislocations) and thus improve the electron mobility in polycrystalline PbTe:In films [16].

## 5. Conclusions

Transport properties (electrical and Hall coefficient) of polycrystalline In-doped PbTe films grown on amorphous (polyimide) substrate by electron gun vapor deposition, were studied over the 10–300 K temperature range at different technological parameters (the substrate temperature, thickness, heat treatment). The unique feature of In impurity in PbTe is the stabilization of the Fermi level (pinning effect), which allowed the preparation of polycrystalline films with the same carrier concentration independent of the technological parameters. We found that heat treatment in an argon atmosphere does not change the average grain size and carrier concentration for as-grown films, but greatly increases the mean free path from 40 nm to 515 nm, which becomes close to the average grain size of 656 ± 50 nm. By comparing the mobility in the bulk and in the film after heat treatment, we extracted the value of the mobility that arises due to scattering at the grain boundary barriers. This mobility has no temperature dependence at low temperatures but has a pronounced thermally activated region close to room temperature. The research on PbTe:In polycrystalline films reported in this paper can pave routes for optimization of the film structure. The ultimate goal of the present study is the development of these films for use in designing portable uncooled photodetectors for the mid-IR range.

## Figures and Tables

**Figure 1 materials-15-08383-f001:**
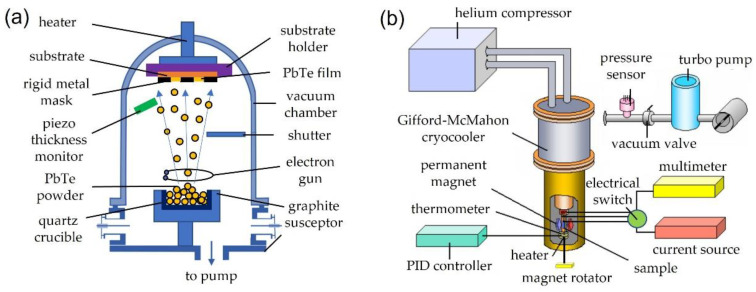
(**a**) The schematic view of the electron gun vapor deposition system. (**b**) Schematic view of set up for measurements of transport properties in PbTe:In films.

**Figure 2 materials-15-08383-f002:**
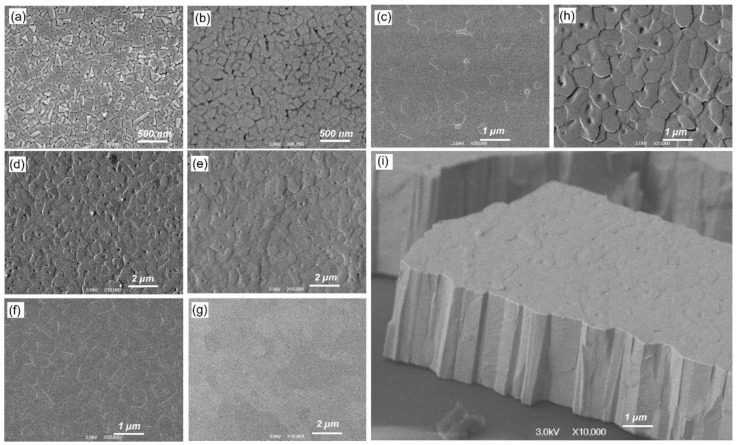
Structural properties of the grown PbTe:In films: (**a**–**c**) SEM image of evolution of the crystallite structure of Pb_0.99_In_0.01_Te films with a common thickness of 2 µm grown on substrates at temperatures 523, 573, 638 K, respectively; (**d**) Grain size for as-grown Pb_0.995_In_0.005_Te film of 3 µm thickness (*T_S_* = 623 K) and (**e**) the same film after heat treatment, respectively; (**f**,**g**) SEM images of evolution of the crystallite structure of Pb_0.99_In_0.01_Te films with 3 µm and 4 µm thickness (*T_S_* = 598 K), respectively; (**h**) Close up of the film surface shown in Figure 2e; (**i**) SEM image of a cross-section of as-grown Pb_0.995_In_0.005_Te film with 3 µm thickness (*T_S_* = 623 K) showing its columnar structure.

**Figure 3 materials-15-08383-f003:**
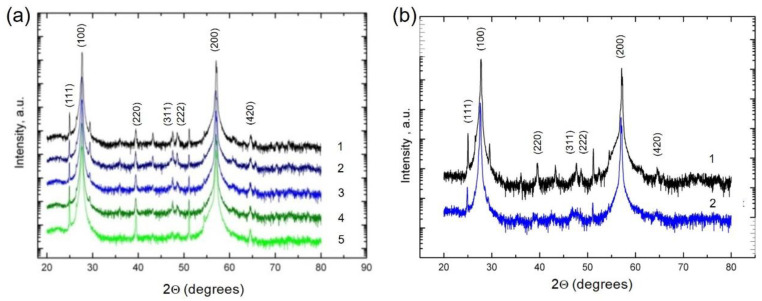
XRD spectra of PbTe:In films on polyimide substrate in a semi logarithmic scale: (**a**) Pb_0.99_In_0.01_Te grown at varied substrate temperatures. Numbers correspond to different substrate temperatures: 1—523 K, 2—548 K, 3—573 K, 4—598 K, 5—638 K; (**b**) Pb_0.995_In_0.005_Te of 3 µm thickness (*T*_S_ = 623 K) before (1) and after (2) heat treatment.

**Figure 4 materials-15-08383-f004:**
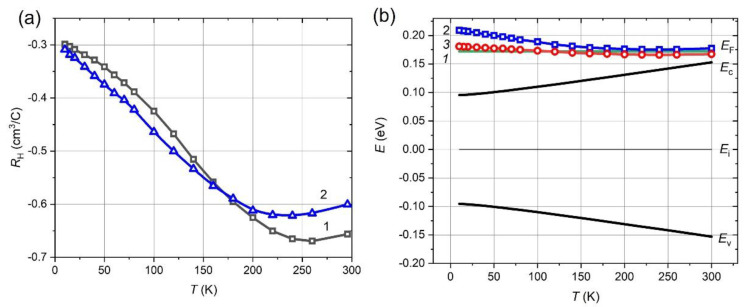
(**a**) Hall coefficient as a function of temperature for as-grown PbTe:In films with different doping level: 1—Pb_0.99_In_0.01_Te, 2—Pb_0.995_In_0.005_Te; (**b**) Reconstructed band energy structure of PbTe on temperature using measured Hall concentration. *E*_V_—valence band ceiling, *E*_C_—bottom of the conduction band, *E*_i_—middle of the bandgap. 1—theoretically predicted Fermi level, 2—calculated Fermi level, using parabolic Kane’s model, 3—calculated Fermi level, using non-parabolic Cohen’s model.

**Figure 5 materials-15-08383-f005:**
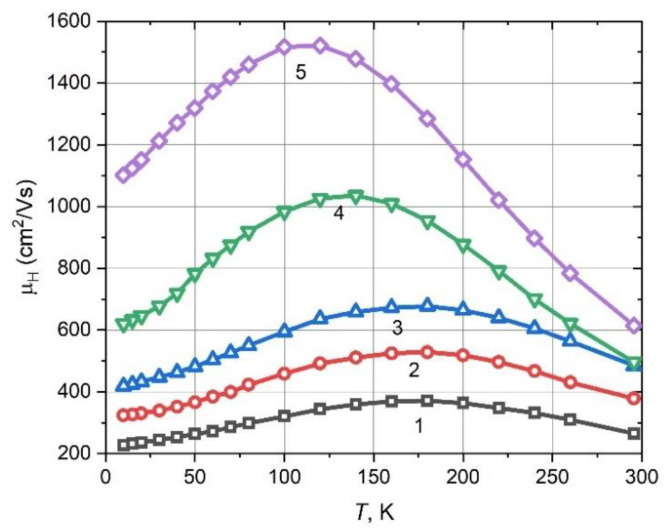
Experimentally obtained temperature dependence of the Hall mobility µ_H_ of Pb_0.99_In_0.01_Te for various substrate temperatures. Numbers correspond to different substrate temperatures: 1—523 K, 2—548 K, 3—573 K, 4—598 K, 5—638 K.

**Figure 6 materials-15-08383-f006:**
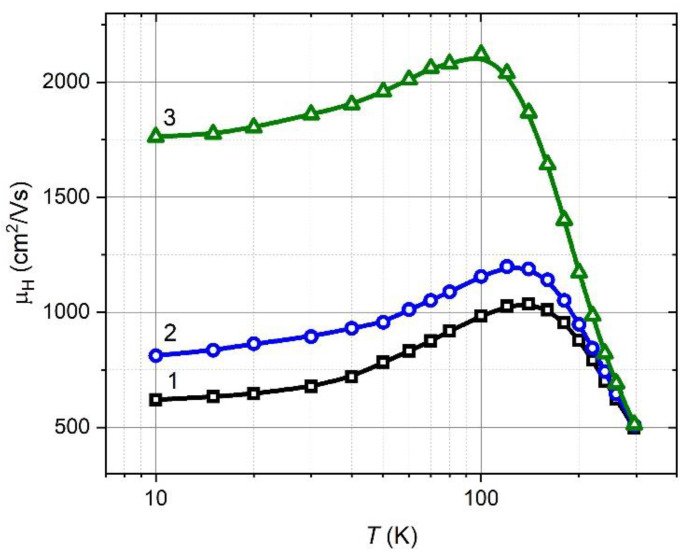
Measured temperature dependence of the Hall mobility µ_H_ of Pb_0.99_In_0.01_Te for various film thickness. Numbers correspond to different film thickness grown *T*s = 598 K: 1–2 µm, 2–3 µm, 3–4 µm.

**Figure 7 materials-15-08383-f007:**
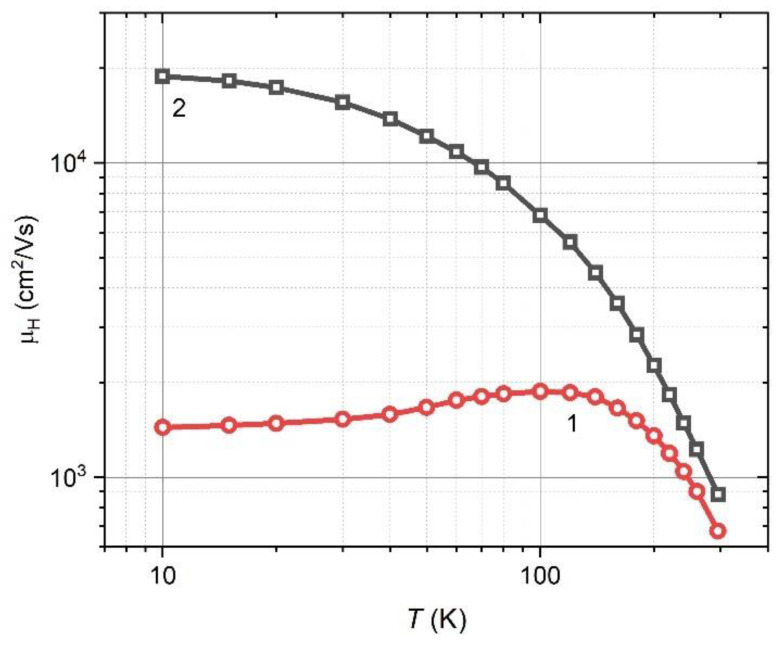
Experimentally obtained temperature dependence Pb_0.995_In_0.005_Te of 3 µm thickness (*T*_S_ = 623 K). Numbers correspond to Hall mobility µ_H_: before—1 and after—2 heat treatment.

**Figure 8 materials-15-08383-f008:**
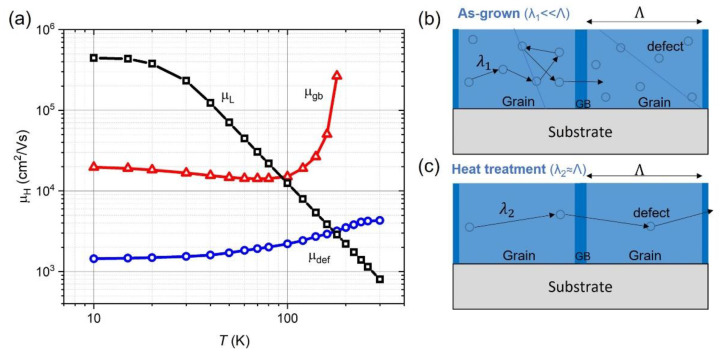
(**a**) Extracted temperature dependence of the electron mobility according to Equation (4). µ_L_—mobility due to lattice scattering in bulk crystal. µ_def_—mobility limited by defects, corresponding to the as-grown film_._ µ_gb_—mobility limited by grain boundary scattering corresponding to the heat-treated film; (**b**,**c**) Schematic views of polycrystalline PbTe:In film before and after heat treatment. λ_1,2_—mean free path for as-grown and after heat treatment, respectively, Λ—grain size.

**Table 1 materials-15-08383-t001:** The main parameters of PbTe:In films on polyimide substrate.

Composition	Thickness, *d* (µm)	Substrate Temperature, *T*_S_ (K)	Average Grain Size (nm)	Mean Free Path at 10 K (nm)	Seebeck Coefficient *S*, (µV/K)
Pb_0.99_In_0.01_Te		523	114 ± 11	6	−160
		548	145 ± 20	9	−150
	2 ± 0.1	573	180 ± 30	11	−160
		598	300 ± 35	18	−150
		638	510 ± 54	29	−150
	3 ± 0.2	598	586 ± 62	25	−160
	4 ± 0.3	598	640 ± 70	44	−150
Pb_0.995_In_0.005_Te	2 ± 0.1	573	180 ± 30	34	−150
	3 ± 0.2	623	641± 70	42	−160
Pb_0.995_In_0.005_Te (heat treatment)	3 ± 0.2	623	656± 50	515	−160

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
