# Peer review of "Polycrystalline PbTe:In Films on Amorphous Substrate: Structure and Physical Properties"

_materials, 2022, doi:10.3390/ma15238383_

Round 1

Reviewer 1 Report

This paper presents a study clearly turned toward applications, but it is supported by a sound scientific argumentation and a good knowledge of the physics of the system. The work has been systematic  and thorough.

I have detected a few minor typos:

line 110: "we used an original the set-up..."

line122: the title from table 1 shoud jump to the next page

line 187: "respectively" is colored in red

line 293: [X], probably a reference was expected there ?

Reviewer 3 Report

The abstract must be rewritten to highlight the findings of the paper. Uncommonly figure 1 is included in the introduction, please remove it and just mention the reference.

Fig.2 needs redesign to include the captions parts into the figure instead of numbers. ex 3 – substrate, 4 – PbTe film...etc.

The Hall measurements and mobility values required more discussion and referring to literatures.

Round 2

Reviewer 2 Report

The quality of the article is significantly improved.

Reviewer 3 Report

The authors addressed all comments, the manuscript can be accepted for publication.